# Diagnostic Impact of Dual-Time PET/CT with ^68^Gallium-PSMA in Prostate Cancer and ^68^Gallium-DOTATOC in Neuroendocrine Tumors

**DOI:** 10.3390/biomedicines11041052

**Published:** 2023-03-29

**Authors:** Damiano Librizzi, Friederike Eilsberger, Stefan Ottenthaler, Ali Ebrahimifard, Markus Luster, Behrooz H. Yousefi

**Affiliations:** Department of Nuclear Medicine, School of Medicine, Philipps University Marburg, 35043 Marburg, Germany

**Keywords:** ^68^gallium, PET/CT, dual-time, PSMA, DOTATOC, prostate cancer, neuroendocrine tumor

## Abstract

Background: The timing of imaging for ^68^gallium (^68^Ga)-PSMA and ^68^Ga-DOTATOC are stated to be around 60 min post-injection (p.i.). In some lesions, late imaging (3–4 h p.i.) showed advantages. The aim of our evaluation was to demonstrate the relevance of an “early” late acquisition. Methods: We retrospectively evaluated 112 patients who underwent ^68^Ga-DOTATOC-PET/CT and 82 patients who underwent ^68^Ga-PSMA-PET/CT. The first scan was acquired 60 min (±15 min) after application. In cases of diagnostic ambiguity, a second scan was performed 30–60 min later. Pathological lesions were analyzed. Results: Almost half of all ^68^Ga-DOTATOC cases and about one-third of all ^68^Ga-PSMA examinations showed a change in findings due to the second acquisition. In total, 45.5% of neuroendocrine tumor (NET) patients and 66.7% of prostate cancer (PCa) patients showed relevant TNM classification changes. For ^68^Ga-PSMA, there were significant increases in sensitivity and specificity from 81.8% to 95.7% and from 66.7% to 100%, respectively. Statistically significant improvements in sensitivity (from 53.3% to 93.3%) and specificity (from 54.6% to 86.4%) were demonstrated for NET patients. Conclusion: Early second images can improve diagnostics with ^68^Ga-DOTATOC and ^68^Ga-PSMA PET/CT.

## 1. Introduction

Many studies have investigated the optimal time interval between tracer application and the acquisition of ^68^Ga-PSMA-PET/CT and ^68^Ga-DOTATOC-PET/CT. The optimal timing of PET/CT imaging for both ^68^Ga-PSMA and ^68^Ga-DOTATOC is stated in international guidelines to be around 60 min post-injection (p.i.) [1,2]. Additionally, the guidelines discuss performing a late acquisition (3 h after application) for ^68^Ga-PSMA-PET/CT. The basis for this discussion is provided by studies that showed that some lesions could be detected exclusively on late ^68^Ga-PSMA-PET/CT images.

For ^68^Ga-DOTATOC-PET/CT, there have also been a few smaller series that support the thesis that a late recording can also be advantageous in this tracer.

The value of the second image results from the pharmacokinetic aspect of the tracers used and physical decay are, in principle, two opposing mechanisms that have to be weighed against each other in the time management of PET/CT.

As the radionuclide constantly decays, the activity remaining in the body steadily decreases. The earlier the acquisition starts, the higher the measurable intensity, the easier it is to detect the remaining tracer and the more sensitive the imaging so that even small foci can potentially be detected. In particular, the relatively short half-life of ^68^gallium (67.6 min) thus forces a rapid approach. On the other hand, the tracer is only gradually internalized in the target tissue, while its concentration in the surrounding tissues decreases in proportion. Thus, a longer period avoids a possible false-positive interpretation of nonspecific foci, hypothetically resulting in higher specificity. An improved target-to-background ratio can be expected.

However, in routine clinical practice, scheduling early (60 min p.i.) and late (3 h p.i.) acquisition is often difficult due to the PET/CT camera workload and patient comfort. In order to take advantage of both acquisition times, dual-time PET/CT is acquired in our department. After a single tracer application, 2 images are acquired with a time interval of approximately 30–60 min. The aim of this study is to present the benefits of an additional “early” late acquisition.

## 2. Materials and Methods

### 2.1. Patients

Included in the retrospective analysis were patients who received dual-time PET/CT with ^68^Ga-DOTATOC or ^68^Ga-PSMA from January 2014 to December 2016 at Marburg University Hospital. A total of 112 dual-time examinations with ^68^Ga-DOTATOC and 82 examinations with ^68^Ga-PSMA were included.

This study was approved by the local ethics committee (AZ.: RS 22/54).

Dual-time PET/CT with ^68^Ga-DOTATOC was performed in 112 patients (13–88 (median: 57) years). In 95 patients (84.8%) with histologically confirmed neuroendocrine tumors (NETs), the aim was staging for metastasis or primary tumor imaging, and, in 17 patients (15.2%), the indication was to confirm the diagnosis of a suspected NET. In 28 patients (25%), MEN syndrome was known, of which 2 patients had not yet developed NETs.

Dual-time PET/CT with ^68^Ga-PMSA was performed in 82 patients with histologically confirmed prostate cancer (PC) (50–82 (median: 69) years). A total of 54 patients (65.9%) had undergone prostatectomy at the time of examination. The most frequent indication for diagnosis in 54 patients (65.9%) was a biochemical recurrence or an inadequate decrease in the PSA value after an intervention with a curative approach. Other indications were the clinical suspicion of recurrence, e.g., bone pain, as well as pretherapeutic staging or follow-up with a palliative treatment approach.

### 2.2. Investigation Protocol

PET/CTs were performed according to the recommendation of the German Society for Nuclear Medicine using a Siemens Biograph 6 True Point. Activities between 180 and 200 MBq of the ^68^Ga-tracers were administered. The first image was acquired 60 min (±15 min) after the application of the radiopharmaceutical. The first scan was immediately evaluated by an experienced nuclear medicine physician. A second PET/CT image of the region in question could be performed 30 to 60 min after the first in cases of ambiguity.

### 2.3. Analysis

The dual-time PET/CT images were retrospectively analyzed by an experienced nuclear medicine specialist, and the number and locations of pathological lesions that could be detected in the first scan were recorded. This is referred to as “report 1” (A1). A maximum of 9 lesions per body region was defined since quantitative differentiation in the case of excessive and partially confluent metastases is, on one hand, technically difficult and, above all, clinically irrelevant. For comparison, the second image, as well as the first findings, resulting from the synopsis of both images were subsequently included, and, again, the number and localization of all spatial lesions with pathologic enhancement were recorded as “report 2” (A2). Congruent with statement 1, the maximum value of 9 lesions per anatomically defined space was observed. As soon as there was a difference between the two statements, the second image was classified as “relevant to findings”. Otherwise, it was classified as “not relevant to findings”.

In prostate cancer, the classification of the anatomical localizations is based on the TNM system. The following findings are relevant:The local findings of the prostate itself or in the prostate bed after prostatectomy;Suspicious regional lymph nodes;Suspicious non-regional lymph nodes;The presence of bone metastases;The presence of other metastases (i.e., abdominal/thoracic/cerebral).The lesions of NETs were grouped as follows:Gastrointestinal (GI) lesions;Liver lesions;Pancreatic lesions;Abdominal lymph node lesions;Other abdominal lesions;Thoracic lesions (including thoracic lymph node lesions);Cerebral lesions;Skeletal lesions.

Where available, the results were compared with the histologic reports as the gold standard to verify the postulated improvement in terms of sensitivity and specificity. If the histology was not available, the further patient history, i.e., the clinical course and further diagnostic procedures, including later PET/CT images, was summarized as a comparative value. If, in sectional imaging, such as CT, MRI, or subsequent PET/CT, a lesion with size progression was detected at the same site that appeared suspicious in A1 or A2 at an interval of at least half a year and, in addition, clinical symptoms matching the tumor disease and site (e.g., local bone pain in skeletal metastases, carcinoid syndrome in NET, and B-symptoms) were detected, the finding was retrospectively considered “definitely positive”. Here, the retrospective reference to the performed biphasic PET/CT scan was crucial. In contrast, a later radiologically or histologically confirmed recurrence diagnosis at another site for which no correlation could be retrospectively determined in the dual-time image was evaluated as “unclear” at the time of the examination since the corresponding focus may have developed later. The complete absence of clinical and radiological evidence of recurrence during the entire period of follow-up (2–5 years) was required for an evaluation as a “definite negative finding” at the time of examination. At least two clinical controls and one new appropriate and negative cross-sectional imaging with a minimum interval of one year were required. In addition, no radiation or chemotherapy was allowed to have taken place during this time in order to exclude false-negative results due to therapeutic success.

## 3. Results

### 3.1. ^68^Ga-DOTATOC-PET/CT

Among 112 NET patients, a total of 527 lesions were detected by A1. In A2, 544 lesions were revealed. In absolute terms, there were 17 more lesions, which corresponded to an increase of 3.2%. The percentage distribution of lesions among different organ systems was almost identical. The most frequent site of manifestation was the liver, with 171 lesions in A1 (32.4%) and 177 lesions in A2 (32.5%). Positive abdominal lymph nodes (Abd Ln) were observed the second most frequently, at 19.5% (A1) and 20.0% (A2). The pancreas, the intestine, the remainder of the abdomen, and the thorax each accounted for about 8–12%. Cerebral manifestations were rare, accounting for less than 2%. An overview of the absolute values and absolute difference is given in Table 1 and Figure 1.

The detailed analysis shows a clear increase in the number of lesions of the pancreas, liver, and abdominal lymph nodes in the late images, minor changes in the lesions of the thorax, skeleton, and remaining abdomen, and a clear reduction in gastrointestinal foci. There were no differences in brain lesions, so this metastatic localization is not included in the figures. The total number of changes in findings was 23 in the pancreas, followed by the intestine with 19 changes and the liver with 18 changes (Table 2).

If these absolute values are put in relation to the total number of lesions of the organs in the first finding (A1), the pancreas stood out further, with a ratio of 46.0%, and the gastrointestinal lesions yielded a ratio of 38.5%. The relative risk of a change in findings was thus significantly increased for the pancreas and GI tract. The liver, at 10.5%, was already below the average of 13.7%. All other organs were below 6%. We additionally plotted these changes to lesions in Figure 2.

### 3.2. Relevance for Patients

In 55 of 112 patients, there was a difference between A1 and A2. The second image was thus relevant to the findings in 49.1% of cases (Figure 3). Of these, additional lesions were detected by the second administration in 30 patients (54.5% of the cases relevant to the findings). In 22 cases (40%), the diagnosed lesion number decreased by at least 1. In 3 patients (5.5%), both an additional lesion was detected, and a previously suspected lesion was classified as nonspecific, which meant that the total lesion number remained the same, but the findings still changed.

Among the 55 cases relevant to the findings, a change in the TNM stage was observed in 25 patients (45.5%) (Table 3). The T stage changed in 10 cases (18.2%), the N stage changed in 1 case (1.8%), and the M stage changed in 14 cases (25.5%). Five examinations resulted in altered organ findings with no effect on the TNM classification. All patients in this group were diagnosed as distantly metastatic in A1 as A2 at different sites. Figure 1 shows an example of a relevant finding in a patient with a suspected recurrence of pancreatic NET.

### 3.3. Sensitivity and Specificity of ^68^Ga-DOTATOC-PET/CT

Of 112 patients, 11 cases (9.8%) had histologic confirmation as the gold standard. In another 56 cases (50%), the synopsis of further imaging and the clinical course over the following 2–5 years could be used as a reference.

It is important for the assessment of the calculated patient-based sensitivities and specificities that they refer to the mutual reference of A1 to A2. For example, if two lesions were detected in A1 but three lesions were detected in A2, the sensitivity and specificity calculation refer to this additional lesion. If all three lesions were detected histologically or confirmed in the clinical course, statement A1 would be evaluated as “false negative”, although the other two lesions were correctly detected as “true positive”.

In the subgroup of histologically verifiable findings, the sensitivity for simple inclusion in A1 was 28.6% and the specificity was 25.0%, with 95% confidence intervals (95% CI) of 8.2–64.1% and 4.6–69.9%, respectively (Table 4).

Dual-time imaging A2, on the other hand, showed a sensitivity of 71.4% (95% CI: 35.9–91.8%) and a specificity of 75.0% (95% CI: 30.1–95.4%), although the significance of these values was relatively low because of the small number of cases, and no statistically significant improvement can be shown (Table 5).

When the calculation was extended to include cases with references elicited by the follow-up, a total of 67 patients (59.8%) could be included. Therefore, the sensitivity for A1 was 53.3% (95% CI: 39.1–67.1%), and the specificity was 54.5% (95% CI: 34.7–73.1%) (Table 6). For A2, a statistically significant increase in sensitivity to 93.3% (95% CI: 82.1–97.7%) and an equally statistically significant improvement in specificity to 86.4% (95% CI: 66.7–95.3%) were calculated in the extended-reference comparison (Table 7). The *p*-values were 0.000051 and 0.047, respectively.

### 3.4. ^68^Ga-PSMA-PET/CT

In 82 PCa patients, a total of 267 lesions were detected in A1 and 265 lesions were detected in A2. The local distribution was almost identical in both cases, which can be seen in Table 8 and Figure 4.

At 44.2%, regional lymph nodes were the most frequent site of manifestation, followed by osseous metastases (approximately 22%) and non-regional lymphogenic metastases (approximately 17%). Local recurrences in the prostate or the prostate bed were present in 7.5% (A1) and 9.1% (A2), respectively. About 30% of these lesions were detected in patients after prostatectomy. Other abdominal lesions were found in about 7% of patients. The proportion of non-osseous and non-lymphatic metastases in the cerebrum and thorax was very low at 0–1.1%.

The changes in findings per organ in the second acquisition are shown in Table 9. In absolute terms, the greatest dynamic was seen in the regional lymph nodes, which accounted for more than half of all changes in the findings (21/38; 55.3%). Relative to the number of lesions, which also accounted for a large proportion (118 foci in A1; 44.2%), only a slightly above average, but a not significantly increased ratio of 17.8% was observed, with an overall ratio of 14.2%.

The next-highest differences in the findings were among the non-regional lymph nodes, with six changes; in the prostate or prostate bed, with four changes; and in the skeleton and abdomen, with three changes each. In relation to the foci detected in A1, the prostate showed the maximum value of 20.0%, followed by the aforementioned regional lymph nodes (17.8%), abdominal lesions (15.8%), non-regional lymph nodes (12.8%), and bones (5.1%). These variations were all non-significant (*p* values: 0.08–1) in relation to the overall ratio (14.2%).

Since there was only one cerebral lesion detected in A1, which was discarded by the second scan, an evaluation would not be representative.

### 3.5. Relevance for Patients

The differences between A1 and A2 of the dual-time PET/CT scans with PSMA are shown in Figure 5. A change in findings with the second scan occurred in 27 patients (32.9%). Of these, a total of 17 additional lesions were detected in 15 cases.

In an additional 15 patients, 20 foci that were initially evaluated as PSMA-positive were classified as nonspecific areas of uptake. Among them, there were three cases with overlaps in which at least one lesion was added and subtracted elsewhere in the synopsis. Of the 27 changes relevant to the findings, 18 cases (66.7%) showed an influence on the TNM stage. Of these, a decrease in the TNM stage was observed in 12 cases (66.7%), and an increase in classification was observed in 6 cases (33.3%). In one special case, the second admission revealed a simultaneous change of the N status to positive with a negation of the M status. In the percentage calculations, this case was included as a reduction because of the clinically decisive M status; in the following individual consideration, it was included in both the N and M changes, which is why the sum of all percentage subsets (Table 10) resulted in a value above 100%. Among all TNM-relevant cases, a change in the T stage was seen in 2 cases (11.1%), an effect on the N status was seen in 10 patients (55.6%), and an effect on the M status was seen in 7 patients (38.9%). In this context, the M status was only downgraded from M1c to M1b in two cases; in all other cases, the difference was actually the general presence of distant metastases, affected lymph nodes, or local findings. Figure 2 shows an example of a relevant finding in a patient with a suspected recurrence of PCA.

### 3.6. Sensitivity and Specificity of ^68^Ga-PSMA-PET/CT

Overall, 11/82 patients had a histopathologic examination available for comparison (13.4%). In this subgroup, a sensitivity of 88.9% (95% CI: 56.5–98.0%) and a specificity of 50% (95% CI: 2.7–97.3%) were calculated for A1 (Table 11).

For A2, there was no change in sensitivity and a non-significant increase in specificity to 100% (95% CI: 19.8%–100%), which was caused by a single case (Table 12).

In a further 23 patients (28.1%), a clinical radiological reference could be elicited by means of follow-up over at least 2 years. When including these patients, A1 achieved a sensitivity of 81.8% (95% CI: 61.5–92.7%) and a specificity of 66.7% (95% CI: 39.1–86.2%) (Table 13).

A2 achieved a significant diagnostic improvement in this group, with a sensitivity of 95.7% (95% CI: 79.0–99.2%) and a specificity of 100% (95% CI: 74.1–100%) (Table 14). However, with *p*-values of 0.11 and 0.10, respectively, these observations were not statistically significant.

## 4. Discussion

The timing of PET/CT scanning remains a topic of debate, balancing clinical information, patient comfort, and economical/logistic aspects. International guidelines state that the optimal timing of PET/CT acquisition for both ^68^Ga-PSMA and ^68^Ga-DOTATOC is around 60 min post-injection. Additionally, performing a late acquisition (3 h after application) is discussed for ^68^Ga-PSMA-PET/CT [1,2,3]. The basis for this discussion is provided by studies revealing some advantages of late imaging. Schmuck et al. showed that for ^68^Ga-PMSA-PET/CT the target-to-background ratio improved significantly over time (*p* < 0.0001). Late imaging at 3 h p.i. exclusively identified 3.4% of all lesions suggestive of recurrent disease [3]. For ^68^Ga-DOTATOC-PET/CT, there are also a few smaller series that support the thesis that a late acquisition can also be advantageous for this tracer, even if this is not reflected in the guidelines. Velikyan et al. showed that with a two-hour post-injection acquisition, the visibility score chosen by the authors was the highest, although the differences were not significant [4]. In a study performed by Nakamoto et al., the authors revealed that the tumor-to-liver ratio (5.9 ± 4.5 versus 6.2 ± 4.6 (*p* < 0.01)) and standardized uptake values (SUV) for hepatic lesions were slightly higher in delayed scanning than in early scanning (26.8 ± 21.2 versus 28.2 ± 21.2 (*p* < 0.01)), which, however, had no impact on the detection rate. When evaluating bone and peritoneal metastases, these lesions also showed slightly higher SUVs in delayed imaging (*p* < 0.05).

Hoffmann et al. were able to show that the SUVmax values for pelvic lymph node metastases were 25% higher (*p* < 0.001) in late PET/CT, which could also provide additional information in individual patients [5].

Even if the existing evidence for ^68^Ga -PET/CT is rather weak, which emphasizes the importance of our study, studies of ^18^fluorine (^18^F)-fluorodeoxyglucose (FDG)-PET/CT are available that demonstrate the input of dual-time scanning. Several research groups have demonstrated the benefits of the dual-time acquisition for FDG-PET/CT for sarcomas, laryngeal carcinomas, cholangiocellular carcinomas, PC, and pediatric malignancies, among others [6,7,8,9,10].

The results of these studies provide support for the inclusion of a second, later acquisition. The greatest expected advantage is primarily that the course of tracer accumulation between the first and second images can also be assessed.

The biphasic approach can lead to an improved informative value of PET/CT by the following:(1)An increase in sensitivity—if a tracer accumulation initially classified as nonspecific in the first image is still clearly visible, it may be more likely to be specific;(2)An increase in specificity—if a lesion is apparently positive at first but cannot be delineated on the second image, actual internalization of the radiotracer into the tissue is unlikely. The finding should thus be described as a nonspecific enhancement.

In ^68^Ga-DOTATOC-PET/CT for the subgroup of histologically verified cases, there was a clear increase in sensitivity from 28.6% (95% CI: 8.2–64.1%) in A1 to 71.4% (95% CI: 35.9–91.8%) in A2. Similarly, there was a significant improvement in specificity from 25% (95% CI: 4.6–69.9%) in A1 to 75% (95% CI: 30.1–95.4%) in A2. Here, however, a valid statistical result and generalizability can hardly be suggested in view of the limited number of cases. With the addition of the clinical–radiological course confirmation group of patients, a total of 67 ^68^Ga-DOTATOC-PET/CT images yielded a sensitivity of 53.3% for A1 (95% CI: 39.1–67.1%) and a specificity of 54.5% (95% CI: 34.7–73.1%). In contrast, a statistically significant improvement could be recorded by the second image, with a sensitivity of 93.3% (95% CI: 82.1–97.7%) and a specificity of 86.4% (95% CI: 66.7–95.3%) (*p* values: 0.000051 and 0.047). The significant improvements in sensitivity and specificity by combining both images thus confirm the usefulness and superiority of the additional examination in this setting. The results of the biphasic examination in this particular patient group are on par with the general appreciation of PET/CT imaging in NET in the literature: in a meta-analysis by Singh et al., sensitivities of 78.3% to 100% and specificities of 83% to 100% were reported, with a pooled sensitivity of 91% (95% confidence interval: 85–94%) and a pooled specificity of 94% (confidence interval: 86–98%) [11].

For ^68^Ga-PSMA-PET/CT, a similar increase related to late acquisition was demonstrated in our study. A1 showed a sensitivity of 81.8% (95% CI: 61.5–92.7%) and a specificity of 66.7% (95% CI: 39.1–86.2%). Improvements in both criteria could be achieved with the second acquisition: the sensitivity of A2 increased to 95.7% (95% CI: 79.0–99.2%), and the specificity increased to 100% (95% CI: 74.1–100%). However, these observations were not statistically significant, although there was a clear trend toward improvement in test performance. The lack of statistical significance was presumably mainly due to the small number of histologically or clinically and radiologically verifiable cases. For ^68^Ga-PSMA-PET/CT in PCa, Skrobek et al. described lesion- and patient-based sensitivities of 65.9% to 94% and specificities of 93% to 100% [12]. Jilg et al. were able to show a sensitivity and specificity of 93.2% and 100% for a main (left/right pelvic and retroperitoneal) region-based analysis and sensitivity and specificity of 81.2% and 99.5% for a subregion-based (common iliac, external iliac, obturator, internal iliac, presacral, aortic bifurcation, aortal, and caval) analysis [13]. Fendler et al. demonstrated a sensitivity of 0.92 on a per-patient analysis and a sensitivity of 0.90 on a per-region analysis [14].

Overall, the quality criteria of A1 are thus lower than those reported in the literature, with sensitivity, in particular, being significantly below the published results. In contrast, the sensitivity and specificity of A2 are on par with the published reference values. The presumed reason for this observation can be found, identically to NET, in a selection bias by filtering out unclear cases since a second image was initiated only if there were ambiguities after the first acquisition.

Whether the diagnostic benefit is, as postulated, mainly generated by the combination of both examinations or whether it results from better image quality due to the longer time interval to the injection of the radiopharmaceutical, as suggested by the increases in the standard uptake value and the target-to-background ratio with late image acquisition that have been described in some publications, remains an open question.

Our study has some limitations. These are the results of a single center. A multicenter study based on these findings should be encouraged to validate the positive data with data from different centers as well as a larger sample. Our results should be confirmed by a study with a higher number of patients to achieve more statistically significant results. In addition, this is a retrospective evaluation with the appropriate restrictions, and a prospective study should be initiated. The possible psychological impact of a second acquisition on the patient should also be considered, as it can either lead to greater insecurity or give the patient the reassuring feeling of being treated by a particularly careful physician. This should be considered when designing a new study.

## 5. Conclusions

A statistically significant diagnostic benefit was demonstrated for the early second acquisition in neuroendocrine tumors in cases of ambiguity compared to single imaging. There was also a clear tendency for a diagnostic benefit in prostate cancer. Our results show the clinical relevance of an early second acquisition that is feasible for everyday clinical practice with a relatively high workload.

## Data Availability

The data that support the findings of this study are available upon reasonable request (S.O.).

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
