# Peer review of "Diagnostic Impact of Dual-Time PET/CT with 68Gallium-PSMA in Prostate Cancer and 68Gallium-DOTATOC in Neuroendocrine Tumors"

_biomedicines, 2023, doi:10.3390/biomedicines11041052_

Round 1

Reviewer 1 Report

Dear Author’s and team,

Thank you for sharing your article with me. I appreciate the effort and time you have put into your research, and I found your study to be very informative and interesting. Your work has provided valuable insights into PSMA PET/CT imaging, and I commend you for your thoroughness in conducting this study.

However, I do have a few comments and suggestions that I believe could improve the quality and impact of your research. Please find my feedback on your work below.

1,Regarding the potential psychological effects of rescanning patients 30-60 minutes after injection, the article did not mention any such effects. However, it may be useful to conduct further research to investigate this possibility.

2,While it is true that Excel is not the most professional tool for creating graphs, it is still a commonly used software for scientific research. Nevertheless, the author may consider using specialized software such as Prism to enhance the quality of the figures.

3,Including PET/CT images and tissue analysis slides would certainly provide more convincing evidence for the study. As you suggested, finding a typical case among the 100 patients and presenting images of that case would be beneficial.

4,The 3.2% difference in lesions observed in A1 may not be statistically significant, and the author should consider addressing this limitation in their discussion. They may also consider conducting further research with a larger sample size to achieve more statistically significant results.

5,The half-life of 68Ga is indeed very short, but it can still be used for imaging purposes since it emits positrons that can be detected by the PET scanner. However, the resolution of the image may not be as high as compared to earlier scans taken closer to the time of injection. The article did not provide information on the quality of the image at the 60-minute interval, so further research may be necessary to address this question.

Author Response

Thank you very much for your time and effort. We hope to have implemented all points well.

1,Regarding the potential psychological effects of rescanning patients 30-60 minutes after injection, the article did not mention any such effects. However, it may be useful to conduct further research to investigate this possibility.

            We thank the Reviewer for this suggestion and agree that performing another acquisition may have psychologic effects on the patient. However, it is also possible that a "closer look" will increase trust in the attending physicians, it may be useful to conduct further research to investigate this possibility.

2,While it is true that Excel is not the most professional tool for creating graphs, it is still a commonly used software for scientific research. Nevertheless, the author may consider using specialized software such as Prism to enhance the quality of the figures.

We understand this criticism and have optimized the graphs.

3,Including PET/CT images and tissue analysis slides would certainly provide more convincing evidence for the study. As you suggested, finding a typical case among the 100 patients and presenting images of that case would be beneficial.

We agree in this point and have added two typical case.

4,The 3.2% difference in lesions observed in A1 may not be statistically significant, and the author should consider addressing this limitation in their discussion. They may also consider conducting further research with a larger sample size to achieve more statistically significant results.

            We thank the Reviewer for this suggestion and added this point to the limitation paragraph.

5,The half-life of 68Ga is indeed very short, but it can still be used for imaging purposes since it emits positrons that can be detected by the PET scanner. However, the resolution of the image may not be as high as compared to earlier scans taken closer to the time of injection. The article did not provide information on the quality of the image at the 60-minute interval, so further research may be necessary to address this question.

We thank the Reviewer for this point but assume that since the guidelines (eg Fendler et al. Joint EANM and SNMMI procedure guideline for prostate cancer imaging) recommend distinctly later images, ours are in an absolutely adequate qualitative state.

Reviewer 2 Report

This study examines the utility of dual time point imaging for evaluation of neuroendocrine tumors (NETs) and prostate cancer. The authors provided convincing evidence that additional PET imaging 30-60 min after the initial image can increase sensitivity and specificity. Some edits are needed to improve the article’s presentation.

Major comment:

1. Please add a paragraph to the Discussion section regarding the study’s limitations.

Minor comments:

2. Abstract, ln 19: Why are confidence intervals provided for sensitivity and specificity in NET patients but not prostate cancer patients?

3. Please check all grammar and spelling. For example, in Figure 1, the title is misspelled. (“Despribution” should be “Distribution,” etc.) In addition, please try to improve the overall organization and flow of the manuscript. The last few sentences of the Discussion section all seem to be related but have been divided into separate paragraphs, making it harder to follow the author's logic.

4. Discussion, ln 368-370: It is not clear what is meant by the sentence: "A general advantage of the biphasic approach compared with PET-CT, which is usually performed in a simple manner, cannot be reliably deduced from the design of this study." It seems this sentence implies that the study cannot show the benefit of dual time point imaging compared to a single acquisition, despite the conclusion stating that additional image acquisition provides relevant diagnostic information. Please clarify.

Author Response

We thank the Reviewer for his time and efforts and we hope to have implemented all points well.

Major comment:

1. Please add a paragraph to the Discussion section regarding the study’s limitations.

We thank the Reviewer for this suggestion and added an appropriate paragraph.

Minor comments:

2. Abstract, ln 19: Why are confidence intervals provided for sensitivity and specificity in NET patients but not prostate cancer patients?

We thank the Reviewer for this point and have taken the confidence intervals completely out of the abstract in order not to overload it too much.

3. Please check all grammar and spelling. For example, in Figure 1, the title is misspelled. (“Despribution” should be “Distribution,” etc.) In addition, please try to improve the overall organization and flow of the manuscript. The last few sentences of the Discussion section all seem to be related but have been divided into separate paragraphs, making it harder to follow the author's logic.

We apologize for individual typing errors and send it to language editing. In addition, we have revised the discussion to a more homogeneous flowing text in response to this suggestion.

4. Discussion, ln 368-370: It is not clear what is meant by the sentence: "A general advantage of the biphasic approach compared with PET-CT, which is usually performed in a simple manner, cannot be reliably deduced from the design of this study." It seems this sentence implies that the study cannot show the benefit of dual time point imaging compared to a single acquisition, despite the conclusion stating that additional image acquisition provides relevant diagnostic information. Please clarify.

We agree with the reviewer that we have not been clear at this point. We wanted to emphasize with the sentence that our work cannot be used as a proxy for other tracers, for example. After careful consideration, we have decided to delete the sentence.

Round 2

Reviewer 2 Report

The authors have satisfactorily addressed all concerns in the new draft. No further comments.